# Physical Movement Habit Formation in Sedentary Office Workers: Protocol Paper

**DOI:** 10.3390/mps5060094

**Published:** 2022-11-25

**Authors:** Kailas Jenkins, Jena Buchan, Ryan E. Rhodes, Kyra Hamilton

**Affiliations:** 1School of Applied Psychology, Griffith University, 176 Messines Ridge Road, Mt Gravatt, QLD 4122, Australia; 2Menzies Health Institute Queensland, Griffith University, Parklands Drive, Gold Coast, QLD 4215, Australia; 3Faculty of Health, Southern Cross University, Gold Coast Campus, Coolangatta, QLD 4225, Australia; 4School of Exercise Science, Physical and Health Education, University of Victoria, 3800 Finnerty Road, Victoria, BC V8P 5C2, Canada; 5Health Sciences Research Institute, University of California, Merced, CA 95343, USA

**Keywords:** sedentary behavior, physical activity, physical movement, habit, intervention

## Abstract

Engaging in physical movement has a number of mental and physical health benefits, and yet 45% of Australia’s population do not meet the recommended guidelines for physical activity. The current study aims to develop an online habit-based intervention designed to reduce sedentary behavior within the workplace, using environmental cues to instigate simple behavioral changes. Participants in this study will include full time office workers who self-report as having a highly sedentary job and work from either a commercial office, home office, or a mixture of both. Participants will complete a habit-based intervention over a four-week period designed to reduce sedentary behavior by increasing habitual responses to simple physical movement behaviors cued by their environment. Analysis will involve mixed methods ANOVAs to test the efficacy of the intervention. A successful intervention will show a reduction in sedentary behavior as a response to habitual simple physical movement behaviors.

## 1. Introduction

In current industrialized societies, health-risk behaviors, such as high levels of sedentary behavior, have been identified as leading causes of disease-related, preventable deaths [1,2,3,4,5,6]. With the increase in technology and decrease in physically demanding jobs, high levels of sedentary behavior and a general disengagement in physical activity are major global health concerns [7,8,9,10,11,12,13]. A particular subset of the population at risk of these concerns are those working in predominately sedentary jobs, such as office workers. Office workers are at particularly high risk for developing conditions related to poor circulation, posture issues, and vision deterioration resulting from computer usage and prolonged sitting [14,15,16,17,18]. As office workers often spend a large proportion of their day engaging in sedentary behavior at work, the workplace has become an opportunistic environment for targeting health-related behaviors [19,20,21,22].

Social cognition theories, such as the theory of planned behavior (TPB–Ajzen [23]), have been applied widely to understand the determinants of sedentary behavior and physical activity [24,25]. Central to many of these models is the premise that intention, which is assumed to be guided by conscious deliberation, is the most proximal predictor of behavior, where the stronger the intention, the more likely the behavior will be acted on [23,24,26,27,28,29,30]. However, it is well documented that the link between intention and behavior is not perfect and that there remains an intention–behavior gap [31]. This gap between intention and behavior highlights a particular concern for interventionalists who aim to develop programs targeted at promoting health behaviors. As those who volunteer for interventions usually report higher intentions, this pragmatic challenge offers a potential explanation to the general lack of long-term effects of behavior change interventions [32,33]. More recently, literature has looked at incorporating constructs that underpin nonconscious processes, such as habit and implicit attitudes, into models predicting sedentary behavior and physical activity and found these more automatic, nonconscious constructs to be important contributors to behavior [34,35,36,37,38,39].

Habit is defined as a psychological construct that is dependent on associations forming between consistent contextual cues and repeated behaviors [40,41,42]. A key feature of habitual actions is that they require less cognitive processing in comparison to intentional behaviors, making the behavior more accessible to perform [43,44]. Research on habit development suggests that it can take anywhere between 18 to 254 days for a behavior to become habitual, dependent on behavioral complexity [45,46]. Physical activity is considered a complex behavior as it requires numerous choices and conducive environmental factors for the behavior to be performed, rendering it more intentional [46,47,48]. For example, to perform physical activity, choices around clothing, transport, and timing need to be considered, along with favorable environmental factors such as the weather; in comparison to simple behaviors such as taking the stairs rather than an escalator when the opportunity presents itself. By focusing on forming habitual responses to cues within the environment that promote physical movement (i.e., stairs), one can increase the amount of movement performed during the day, thus limiting sedentary behavior without taxing cognitive load [49]. Furthermore, as simple behaviors are more conducive to habit formation, they also require less time to reach habit plateau [46]. This suggests that interventions looking to utilize habit formation towards simple behaviors should highlight the measurement of habit development during the early stages of the intervention [46]. As time constraints are often an influencing factor for attrition in physical activity interventions, this structure offers a unique method to counter this barrier [19,22,50].

Studies looking at developing interventions that are aimed at developing habits around performing physical movement should focus on four key aspects: the target behavior is personally relevant; the target behavior is considered a simple change to implement within pre-existing practices; the target behavior is realistic to action; and the target behavior can be monitored and assessed against behavioral goals [45,51]. Furthermore, when developing interventions, previous literature has looked at implementation techniques, comparing internet-based to in person delivery modes. While there appears to be advantages and disadvantages for each method, an internet-based delivery mode offers potential methods for countering common barriers associated with physical activity interventions, such as time constraints. As these barriers contribute to poor attrition rates within physical activity interventions, online interventions provide flexible delivery modes with the added opportunity to reach wider populations [52,53]. Hamilton, Fraser [49] conducted an online pilot study aiming to develop habits around simple physical activity behaviors (e.g., moving the wastepaper bin further away from the desk to encourage more steps) within the workplace, utilizing features outlined by Gardner et al. (2014, 2020). This study found positive trends for habit development among sedentary office workers over a six-week intervention, providing preliminary support for this model.

The current study aims to develop an online habit-based intervention designed to reduce sedentary behavior within the workplace, using environmental cues to instigate simple behavioral changes. Four hypotheses are preregistered for the current study. Based on previous research, the primary objective is to test the effectiveness of the intervention on limiting occupational sedentary behavior and increasing occupational physical movement habits within the workplace. It is hypothesized that reported occupational sedentary behavior will have a decrease (H1) and reported occupational physical movement will have an increase (H2) across baseline (T1) and follow up points two weeks post baseline (T2) and four weeks post baseline (T3) in comparison to those in the control group. It is further expected that occupational physical movement habit will have an increase (H3) and occupational sedentary behavior habit will have a decrease (H4) across baseline (T1) and follow up points two weeks post baseline (T2) and four weeks post baseline (T3).

## 2. Experimental Design

The study is a randomized control trial, designed to reduce occupational sedentary behavior by forming habits towards physical movement within the workplace environment using an online intervention. The study is guided by the SPIRIT (Standard Protocol Items: Recommendations for Interventional Trials) statement [54] (see Appendix B and Appendix C). Participants are recruited from Australian Universities and the general public via email broadcast and provide informed written consent upon registration of their interest to participate. The intervention will be implemented online using Qualtrics software. After baseline data is collected, participants are randomly assigned to either the (1) control group or (2) intervention group electronically through the randomization function within Qualtrics software. Participants will be blinded to the study condition they are assigned to. Time points involve participants completing online surveys assessing social psychological and behavioral constructs at Time 1 (baseline; T1), Time 2 (two weeks postbaseline; T2), and Time 3 (four weeks post baseline; T3), see Appendix D. The intervention protocol is documented and registered on Open Sciences Framework (OSF) and will be updated upon any deviations (see Appendix A). Full ethical approval was granted by the Griffith University Human Research Ethics Committee on 11 June 2020 and last updated on the 8 July 2022 (GU: 2020/401). Any adverse events that occur during the intervention will be documented and reported to the University ethics committee. All data are stored in a secure database accessible to the research team only, in accordance with Griffith University Research Ethics Committee protocol (GU: 2020/401). Data will be checked for errors and deidentified upon collection at T3, with all analyses conducted based on the principle of intention-to-treat [55]. The final data set will be analyzed, with findings written up for publication in peer-reviewed journals and industry reports and for presentation at scientific conferences, and the deidentified data made available on the OSF platform.

## 3. Procedure

### 3.1. Recruitment Procedures and Inclusion/Exclusion Criteria

Participants are recruited via the Griffith University email broadcast, University networks, and through social media advertisement. Participants are included if they are aged 18 years or older; self-describe as having a highly sedentary job (i.e., sit for at least 75% of the working day); and work full-time from either a commercial office, home office, or a combination of the two. Participants are excluded if they self-describe as not wanting to limit their sedentary behavior at work.

### 3.2. Sample Size

To test intervention effects, there is an intended recruitment of 226 participants. Based on previous research in this field and ongoing research within this population group [56], it is anticipated that there will be approximately a 25% attrition over the four weeks of follow-up for reasons such as changes in job status, vacancy, and failure to complete questionnaires. A total sample of approximately 180 completing participants (90/group) is required to detect a medium effect in habit development towards physical movement. This sample was calculated by a power analysis using WebPower program [57]. Significance level (alpha) was established at 0.05 to avoid a Type 1 error and effect size was determined at f = 0.35. This means that for a 95% chance of detecting a significant intervention effect at a four-week follow-up, approximately 90 participants are needed in each group.

### 3.3. Measurement

Data will be collected online via self-reported pre- and post-intervention questionnaires hosted on Qualtrics software. Baseline (T1) data will be collected upon registration for the intervention, T2 data will be collected two weeks post-baseline, and T3 data will be collected four weeks post-baseline (Figure 1).

### 3.4. Demographic Measures

Demographic data will be collected at baseline and will include age (in years), gender (male, female, non-binary, and prefer not to say), residential postcode, current marital status (married registered, de-facto relationship, widowed, divorced, separated, never married), workplace environment (commercial office, home office, mixture of commercial and home office), occupation, education level, family taxable income range, and ethnicity.

### 3.5. Measured Variables

The primary outcome variables will assess the effectiveness of the intervention on limiting occupational sedentary behavior and developing habits towards occupational physical movement within the workplace. The Occupational Sitting and Physical Activity Questionnaire (OSPAQ) is used to measure occupational sedentary behavior and occupational physical movement. The OSPAQ has been validated for use with both commercial office and home office working populations [58,59,60]. To measure the construct of habit for occupational sedentary behavior and physical movement the four-item Self-Report Behavioral Automaticity Index is used [61]. For occupational physical movement habit, participants will be asked to rate their agreement with the following statements “Do you agree that doing occupational physical movement as part of your daily work routine is something: I do automatically; I do without having to consciously remember; I do without thinking; I start doing before I realize I am doing it”, measured on a 7-point Likert scale (1 = strongly disagree, 7 = strongly agree). To measure habit to engage in occupational sedentary behavior, participants will be asked to rate their agreement with the following statements “ Do you agree that engaging in occupational sedentary behavior as part of your daily work routine is something: I do automatically; I do without having to consciously remember; I do without thinking; I start doing before I realize I am doing it”, measured on a 7-point Likert scale (1 = strongly disagree, 7 = strongly agree). Similar items have been used in previous research [23,49,62].

### 3.6. Study Conditions

#### 3.6.1. Control Group

Participants in the control group are provided with an information sheet adapted from the World Health Organization recommendations on physical activity as well as the Canadian guidelines on sedentary behavior [63]. The World Health Organization recommends adults should aim to do more than 150 min of moderate to vigorous physical activity over the week [13]. Canadian guidelines for sedentary behavior involve limiting sedentary time to eight hours or less and breaking up long periods of sitting as often as possible, along with replacing sedentary behavior with additional physical activity [63]. The information pertains to the risks of increased sedentary behavior and the health benefits of increased movement during the day.

#### 3.6.2. Intervention Group

Along with the information sheet, participants allocated to the intervention group will be provided with a digital poster outlining the positive normative beliefs of other office workers towards movement in the workplace. This poster is developed based on previous literature [49] and guided by literature on the social norms approach [25,64,65,66,67]. Along with this poster, participants are provided with the ‘10 Top Tips’ poster, which includes 10 simple behaviors that can be performed within the workplace to increase movement during the day. Participants are prompted to choose tips that they are confident they could perform and write an action plan outlining where, when, and how they intend to implement their chosen tips, along with a coping plan of how they intend to overcome potential barriers to behavioral production. Following this, participants will be asked to write an encouraging statement to instigate their self-efficacy towards achieving their tips. Furthermore, participants will be provided with a self-monitoring tick sheet that can be used to monitor their behavioral progress. The intervention group is encouraged to print their action and coping plans as reminders of their intentions and to email the self-monitoring tick sheet to the first author at the end of each week.

## 4. Data Analyses

To assess the effect of the intervention on reducing occupational sedentary behavior and increasing occupational physical movement; as well as the interventions effect on increasing occupational physical movement habit and its effects on decreasing occupational sedentary behavior habit, four mixed methods 2 (condition/group) × 3 (time) ANOVA’s will be conducted. To assess the overall effects of the intervention, group (control or intervention) will be used as the between-subjects independent variable; time (baseline/T1, T2 and T3) will be used as the within-subjects independent variable with occupational sedentary behavior, occupational physical movement, occupational physical movement habit, and occupational sedentary behavior habit as the dependent variables. Each ANOVA will use an adjusted alpha level of 0.0125 to protect from inflation of type I error. Results, however, will report on any effects using the conventionally accepted significance cut-off level of 0.05 as well as the pre-specified cut-off level of 0.0125. Where an ANOVA indicates a significant time*group interaction for either of the dependent variables, a simple effects analysis will be conducted.

## 5. Expected Results

This study trials a workplace-based intervention aiming to reduce occupational sedentary behavior and increase occupational physical movement through developing habits towards increasing simple physical activity behaviors throughout the workday. To date, limited interventions have used habit formation theory to reduce sedentary behavior and promote physical movement within workplace settings [68]. This theory-driven, multi-component intervention aims to address this gap by providing simple behavioral changes to increase the level of general, opportunistic movement within a sedentary population. A successful intervention will reduce sedentary behavior by developing habitual responses to simple physical activity behaviors, lessening the cognitive demand related to complex physical activity [46]. The successful intervention will be easily translatable to the general population and may be modified to suit other highly sedentary populations.

## Figures and Tables

**Figure 1 mps-05-00094-f001:**
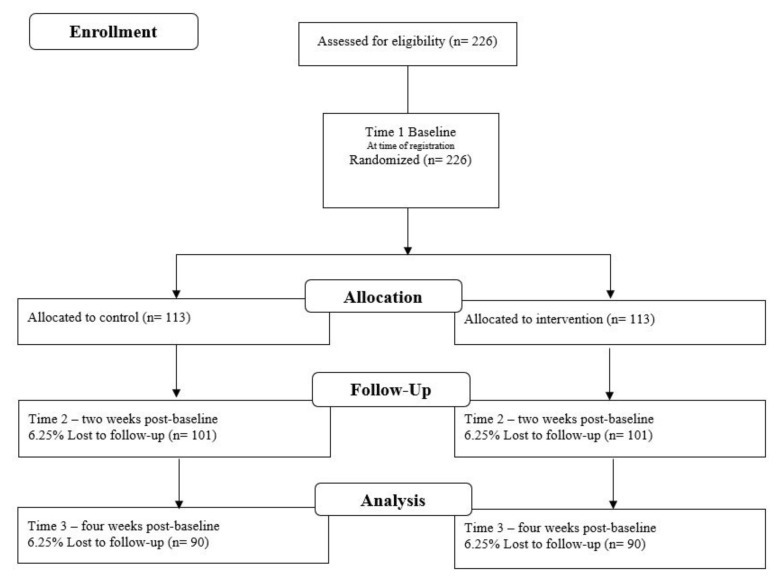
Flow of Participants Through the Intervention.

## Data Availability

Results will be written up and peer reviewed for publication. The study is preregistered on the Open Science Framework platform (OSF, registration number: DOI 10.17605/OSF.IO/JASTP).

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
