# Peer review of "Physical Movement Habit Formation in Sedentary Office Workers: Protocol Paper"

_mps, 2022, doi:10.3390/mps5060094_

Round 1

Reviewer 1 Report

The article explores an interesting question about the reduction of  sedentary behavior by increasing habitual responses to simple physical movement behaviors cued by their environment. The research process is adequate, but there is no relevant content for the discussion of key data and problems. The conclusion and results of the study are not clear to the reader.

Author Response

Thank you for your review, please see attached responses. 

Kind regards,

Kailas Jenkins

Reviewer 2 Report

I read with interest the protocol paper titled "Physical Movement Habit Formation within Office Workers: Protocol Paper". The manuscript is clear and well-justified. Also, the methods are correct. Please, read the following suggestions constructively. 

Only I have two minor comments:

  • Why do the authors use a p-value of 0.01 instead of the Bonferroni correction?
  • A 25% drop-out could reveal that the intervention does not have much adherence.

Author Response

(The authors gave the same response as above.)
